# Generation of Decoy Signals Using Optical Amplifiers for a Plug-and-Play Quantum Key Distribution System

**Min-Ki Woo** [1], **Chang-Hoon Park** [1,2], **Sangin Kim** [1] and **Sang-Wook Han** [2,3,*]

1 Department of Electrical and Computer Engineering, Ajou University, Suwon 16499, Korea; namdo6sung@hanmail.net (M.-K.W.); originalpch@kist.re.kr (C.-H.P.); sangin@ajou.ac.kr (S.K.)
2 Center for Quantum Information, Korea Institute of Science and Technology (KIST), Seoul 02792, Korea
3 Division of Nano & Information Technology, KIST School, Korea University of Science and Technology, Seoul 02792, Korea
* Correspondence: swhan@kist.re.kr

**Abstract:** In most quantum key distribution (QKD) systems, a decoy-state protocol is implemented for preventing potential quantum attacks and higher mean photon rates. An optical intensity modulator attenuating the signal intensity is used to implement it in a QKD system adopting a one-way architecture. However, in the case of the plug-and-play (or two-way) architecture, there are technical issues, including random polarization of the input signal pulse and long-term stability. In this study, we propose a method for generating decoy pulses through amplification using an optical amplifier. The proposed scheme operates regardless of the input signal polarization. In addition, a circulator was added to adjust the signal intensity when the signal enters the input and exits the QKD transmitter by monitoring the intensity of the output signal pulse. It also helps to defend against Trojan horse attacks. A test setup for the proof-of-principle experiment was implemented and tested, and it was shown that the system operated stably with a quantum bit error rate (QBER) value of less than 5% over 26 h using a quantum channel (QC) of 25 km.

**Keywords:** plug-and-play quantum key distribution; decoy-state protocol; quantum optics; quantum communication; QKD; information security

## 1. Introduction

Quantum key distribution (QKD) is a mature technology widely known to be the closest to commercialization among quantum application technologies [1]. QKD allows two authenticated distant participants, Alice and Bob, to share quantum secret keys securely based on quantum effects [2]. It has evolved over the decades since the first BB84 protocol was proposed [3], and the first commercial product appeared in 2007. To achieve a more prevalent technology, many researchers are still conducting long-distance transmission (optical fiber, satellite) [4,5], network [6], chip-based [7], and long-term stability tests in actual testbeds implemented in the local area [8–10].

More specifically, in terms of commercialization, system stability is one of the key issues in QKD systems. Essentially, a quantum signal is fragile to changes in the external environment, and a delicate control is required for long-term stable operation. From this point of view, plug-and-play (PnP) QKD [11], proposed in 1997, has the advantage that a stable operation is possible even if environmental conditions change abruptly. Therefore, it has a round-trip structure. However, the PnP system still has some technical issues such as a relatively low secret key rate and the need for a bulky storage line. More than anything, the implementation of the decoy-state protocol [12–17] is relatively tricky, unlike the conventional one-way QKD architecture. The decoy-state protocol was originally proposed to overcome photon splitting attacks. It allows QKD to operate with a higher average photon number, which increases the quantum key generation rate. In QKD using a decoy-state, the intensity of each pulse should be adjusted corresponding to a random

number that determines its position and intensity. For adjusting pulse intensity, an intensity modulator (IM) is typically used. In a conventional one-way QKD, there is no issue with using a polarization-dependent IM because it can be implemented to use only a specific polarization within the transmitter (or Alice) system. However, in the case of PnP, it is impossible to implement with the simple connection of IM because the signal delivered from Bob to Alice has random polarization due to the quantum channel (QC) comprising single-mode fibers. If QC is implemented with polarization-maintaining fibers, this issue may disappear, but for cost reasons, almost all of the currently installed fibers are composed of single-mode-fibers. Therefore, for resolving this issue, several technologies have been proposed, including the use of a polarization beam splitter (PBS) [18,19] and the control of the $V_{bias}$ value [20,21]. However, stability remains a problem, and it requires an additional separate feedback control. It is not easy to control by monitoring the output signal because it handles very tiny signals.

We propose a structure that controls the intensity of light based on an optical amplifier (OA). This structure generates decoy pulses by controlling the amplification rather than the attenuation of pulses received from Bob. The OA used for amplification is highly stable in terms of signal output, resulting in little variation in the output signal during long-term operation. Thus, it is possible to generate a stable and reliable signal and decoy pulse.

## 2. PnP QKD Decoy System with Amplification Structure

The proposed structure was implemented as shown in Figure 1. The main flow of the pulse is as follows:

① Bob generates a strong laser pulse (1550.92 nm) with a gain-switched laser. Then, the pulse passes through the asymmetric Mach–Zehnder interferometer and is divided into a time-bin pulse $|f\rangle + |s\rangle$ and transmitted to Alice. Now, in our scheme, $|f\rangle$ has horizontal polarization, while $|s\rangle$ has vertical polarization.

② After passing through the QC, some part of the strong pulse that reaches Alice branches to ⑥. The rest goes to an attenuator (Att_1) forward-directed circulator (Cir), storage line (SL), bandpass filter, and OA.

③ The OA amplifies the intensity of the pulse. Noise from the OA is reduced by a bandpass filter. Subsequently arriving at the phase modulator (PM), which is used for phase encoding and randomization, the pulse is then attenuated at Att_2, reflected from the Faraday mirror (FM), and proceeded backward to OA. Then, it is amplified once again in OA.

④ The pulse proceeds in reverse order of ②. At this point, most of the pulse power is delivered in the circulator's forward direction, ⑦. The only rest weak power of the pulse is transmitted to att_1, and it is fine-tuned to a single-photon level at Alice's output.

⑤ The single-photon level pulse passes through the QC and reaches Bob's PM. Decoding starts through this PM. After passing Bob's asymmetric Mach–Zehnder interferometer, it reaches the beam splitter (BS) and interferes. The results of the interference can be known from the measurement results of the single-photon detectors (SPDs).

⑥ A pulse that splits from the BS of Alice reaches photo diode_1 (PD_1). This pulse enables the timing control of Alice's device. Moreover, measuring the intensity of the pulse may help to adjust the intensity of Bob's initial laser pulse.

⑦ This is the forward flow of the circulator. It is connected to PD_2, and the output pulse can be monitored through PD_2.

The intensity of each pulse for the decoy-state protocol can be controlled by driving the OA component. In this case, the control of the OA is achieved with simple open-loop control without complicated feedback control. In addition, the OA control timing is easily adjusted using the time detected by PD_1. The ideal OA gives the desired amplification regardless of the characteristics of the input pulses. However, in reality, a semiconductor optical amplifier (SOA) has a few technical issues that need to be addressed. These are from a device imperfection.

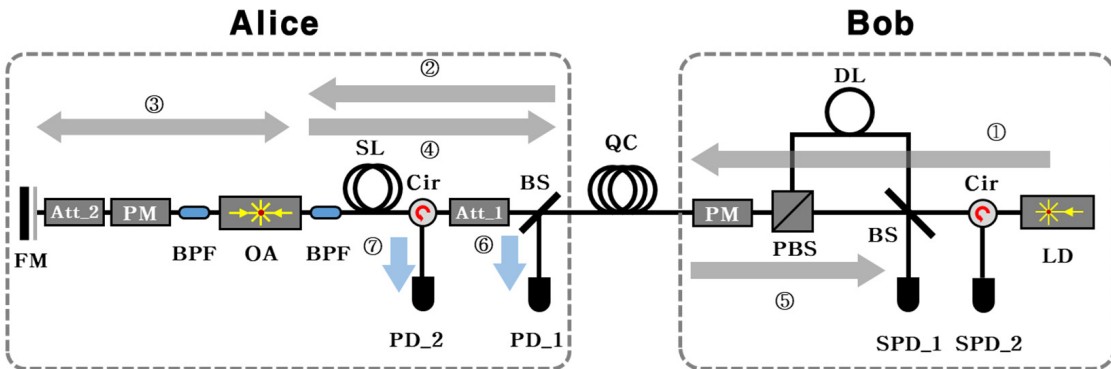

**Figure 1.** PnP QKD architecture with decoy system. The overall structure is similar to the original PnP QKD, but an optical amplifier (OA), band pass filters (BPFs), a circulator, and an attenuator have been added for generating decoy pulses in Alice. An initial pulse is generated in Bob's laser diode and then passed through an asymmetric Mach–Zehnder interferometer to generate a time-bin pulse. At this time, to create a similar intensity of pulses of $|f\rangle$ and $|s\rangle$, the phase modulator in Bob is placed behind the polarization beam splitter. Amplification-based decoy states are generated at OA using double amplification method. The circulator in Alice is used to monitor the output signal, and it plays a role in large attenuation of the pulse passing in the reverse direction of the circulator, as well. After that, the pulse is precisely attenuated to a single-photon level by the attenuator of Att_1 and transmitted to Bob. (FM: Faraday mirror (NFRM-15, General photonics), PM: phase modulator (PM-0S5, Eospace), BPF: Bandpass filter (of003, Fiberpia), OA: optical amplifier (SOA1117P, Thorlabs), SL: storage line, Cir: circulator (FPCR-155, Haphit), BS: beam splitter (FPCL-1550, Haphit), PD: photodiode (TPA-4NN3, Teradian), Att: attenuator (bc, Fiberpia), QC: quantum channel (single-mode fiber), PBS: polarization beam splitter (PBS-1550, AC photonics), DL: delay line (polarization-maintaining fiber), SPD: single-photon detector (SPD_a_M2, Aurea), LD: laser diode (FRL 15DCWA-A81-19330, Fitel [1550.92 nm])).

First, the pulse intensity to reach an OA must be less than $-25$ dBm. If this is not the case due to an output pulse saturation problem on the typical OA, the amplification rate drops abruptly. It is difficult to control the intensity without distorting the pulse if the pulse is amplified in the range where gain drops abruptly (input intensity $> -25$ dBm). By adjusting the intensity of Bob's initial transmission pulse, or the attenuation rate of Att_1, a pulse less than $-25$ dBm is input to the OA.

Second, as shown in Figure 2, OA still has a polarization dependence characteristic. A single-mode fiber, which is often used as a QC, does not maintain pulse polarization. Therefore, the pulse transmitted from Bob to Alice becomes randomly polarized. Typically, amplifying the pulse with random polarization causes pulse distortion because the amplifier's gain in the vertical and horizontal components is different. To solve the polarization issue, we installed FM and Att_2 and used the double amplification method. FM reflects the phase of the pulse orthogonally, and Att_2 makes the pulse become less than $-25$ dBm. A pulse with randomly polarization is expressed as the vertical and horizontal components as follows:

$$\text{Pulse} = A|V\rangle + B|H\rangle, \ |A|^2 + |B|^2 = 1 \tag{1}$$

where *A* and *B* are the intensities of each component. When this pulse was amplified at the initial OA, the vertical and horizontal components were amplified at amplification rates of $G_V$ and $G_H$, respectively.

$$\text{Pulse} = (A \times G_V)|V\rangle + (B \times G_H)|H\rangle. \tag{2}$$

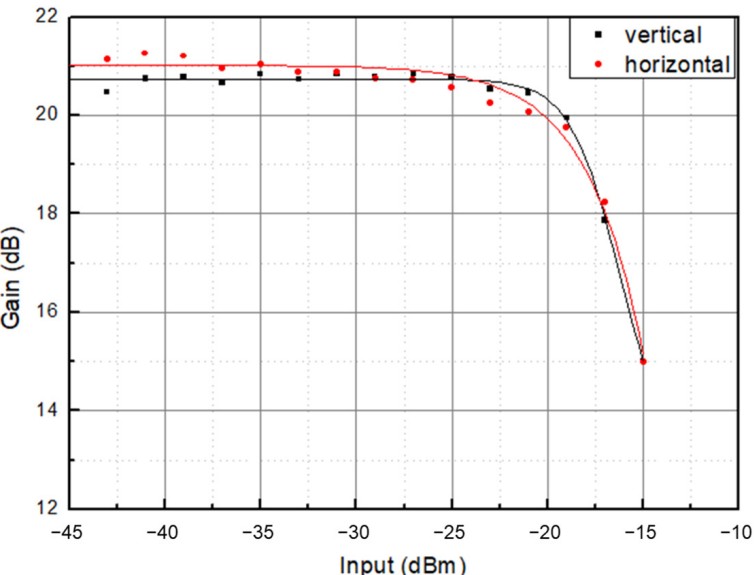

**Figure 2.** Gain of OA according to polarization and optical input power.

The amplified pulse was reflected from FM, and the polarization changed orthogonally.

$$\text{Pulse} = (A \times G_V)|H\rangle + (B \times G_H)|V\rangle. \tag{3}$$

If the pulse was amplified in OA again,

$$\text{Pulse} = (A \times G_V \times G_H)|H\rangle + (B \times G_H \times G_V)|V\rangle. \tag{4}$$

The results of Equation (4) show that both the H and V components are amplified at the same amplification rate, $G_V \times G_H$. When amplified using the proposed structure, no pulse distortion occurred. As long as the output of the OA is not saturated, the double amplification method works well.

Third, the pulse reaching OA must be larger than −50 dBm. During operation, the OA generates noise by itself to the level of −30 dBm. Considering the gain (approximately 20 dBm), the intensity of the input pulse must be greater than −50 dBm to distinguish pulse from noise.

Thus, three technical issues that may arise due to the characteristics of the OA itself can be resolved successfully. However, there is still one more technical difficulty when implementing attenuation. Because the pulse is amplified at OA in the proposed structure, much more attenuation is needed to make a single-photon-level pulse. If large attenuation is implemented with a typical passive attenuator for lowering the intensity to the single-photon level, it attenuates the input pulse of Alice as well; therefore, the intensity of the pulse at OA may drop below −50 dBm. On the other hand, unlike the general PnP QKD, our proposed structure uses a circulator as the main attenuator. In the pulse path of ② in Figure 1, the input pulse is not attenuated, and only the output pulse in the pulse path of ④ is significantly attenuated (around 40 dBm). With the aid of Att_1, an accurate attenuation rate for generating single-photon-level pulses is precisely adjusted. In addition, the OA output pulse proceeding in the forward direction of the circulator (⑦) can be easily measured with a general photodiode. This makes it easy to monitor the output pulse, which was difficult in the conventional PnP-type structure, and by using it, it can also defend efficiently against Eve's Trojan horse attack [22,23].

### 3. Method and Results

An experiment was conducted to verify the feasibility of the structure described in Figure 1. First, the characteristics of the OA to be used were experimentally verified. We used an SOA of SOA1117P (Thorlabs) as an OA. For supplying an input current of 400 mA to the SOA, an SOA driver of SOA-std (Aerodiode) was used. After that, we set the laser of FRL 15DCWA-A81-19330 (Fitel) to have vertical polarization, and it was inputted to the OA while changing the power. The amplified output pulse from the OA was measured with an optical power meter of PM100D (Thorlabs). Afterward, the same experiment was repeated by setting the laser to horizontal polarization.

Next, the double amplification method was experimentally verified. Experiments were performed with laser pulses that had a pulse width of 3 ns. The pulses input to the OA were time-bin pulses $|f\rangle$ and $|s\rangle$. $|f\rangle$ is a pulse that passes through a short path (BS-PBS) among the divided pulses in BS, and $|s\rangle$ is that which passes through a long path (BS-delay line (DL)-PBS) in Bob's system, as shown in Figure 1. The time interval between time-bin pulses $|f\rangle$ and $|s\rangle$ is 100 ns. In PD_1 of TPA-4NN3 (Teradian), the original time-bin pulses $|f\rangle$ and $|s\rangle$ are measured. We measured the output value of the PD using an oscilloscope of RTM 3K (Rohde and Schwarz). At this time, intensity of $|s\rangle$ is subtly smaller than $|f\rangle$ owing to the insertion loss of DL. In PD_2, $|f\rangle$ and $|s\rangle$ of the amplified pulse are measured. When there is no difference in amplification rate due to polarization, the intensity ratio of $|f\rangle$ and $|s\rangle$ of the original pulse and the amplified pulse becomes the same. It is important to confirm that the amplification rate of the OA is reproducibly constant. For verifying reproducibility, we measured 10,000 times.

Next, an experiment was conducted to adjust the intensity of the pulse by controlling the OA input current. We adjusted the OA input current from 350 to 450 mA and verified whether we could produce a pulse (signal (average photon number: 0.5), decoy (average photon number: 0.1), vacuum pulse) of the desired intensity.

The interference visibility of the MZ interferometer formed by Bob and Alice was measured using an amplified pulse. Except for the vacuum state, it was measured using the signal and decoy pulses. The interference visibility can be obtained from the result of a pair of SPD of SPD_A_M2 (Aurea) measurements of Bob according to the change in the input pulse of Alice's PM of PM-0S5-12 (Eospace).

Finally, we measured the key generation rate and QBER. Except for the QC, we implemented an experimental setup, as shown in Figure 1. QC was replaced with a passive attenuator (5 dB), which has the same attenuation rate as QC (25 km, single-mode fiber, 0.2 dB/km (1550 nm)). As in the previous experiment, the width of the generated laser pulse was 3 ns, and the interval between $|f\rangle$ and $|s\rangle$ was 100 ns. A single-photon was detected with an SPD using gate mode. The detector had a gate width of 4 ns and a detection efficiency of 15%. The OA was controlled through a simple open-loop control to generate signal pulses and decoy pulses. We conducted static modulation for signal and decoy pulse intensities.

The measurement results of the change in the OA's amplification rate according to the input pulse intensity and polarization are shown in Figure 2. As mentioned earlier, OA's gain drops abruptly when the pulse power input is higher than −25 dBm. Additionally, the amplification gain is different depending on the polarization.

The double amplification method experiment results are shown in Figure 3. The results show that the original pulses have an intensity ratio of 1:0.82 and the amplification pulses have an intensity ratio of 1:0.82. The results show that the double amplification method amplifies pulse intensity regardless of its polarization.

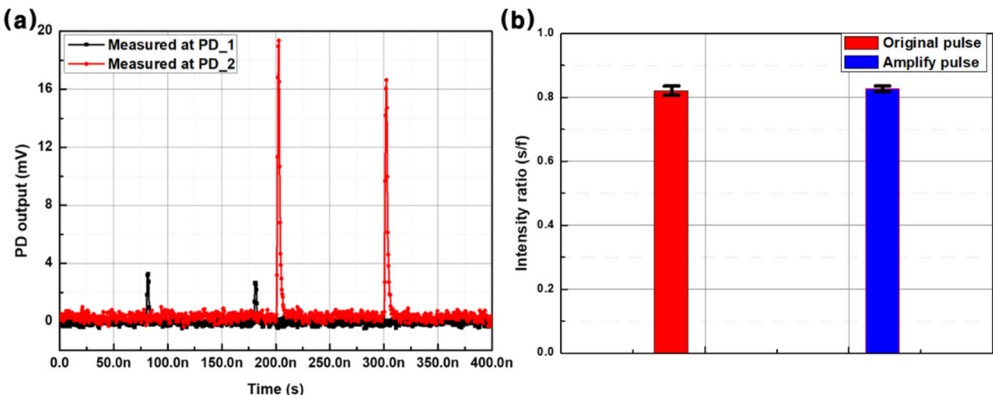

**Figure 3.** Double amplification method results: (**a**) intensity of the source and amplification pulses of $|f\rangle$ and $|s\rangle$, (**b**) comparison of $|s\rangle/|f\rangle$ values of original pulse and amplitude pulse.

The experimental results for controlling the intensity of the pulse are shown in Figure 4. The signal, decoy, and vacuum state can be generated by driving the OA input currents at 350, 420, and 0 mA, respectively. In particular, when OA was in the off state (=0 mA), the extinction ratio reached 46 dB, and thus, the vacuum state can be generated more reliably than a commercial IM of A2-0S5-10 (Eospace) (>20 dB). This result shows that pulses with different average photon numbers for the decoy-state protocol can be generated well.

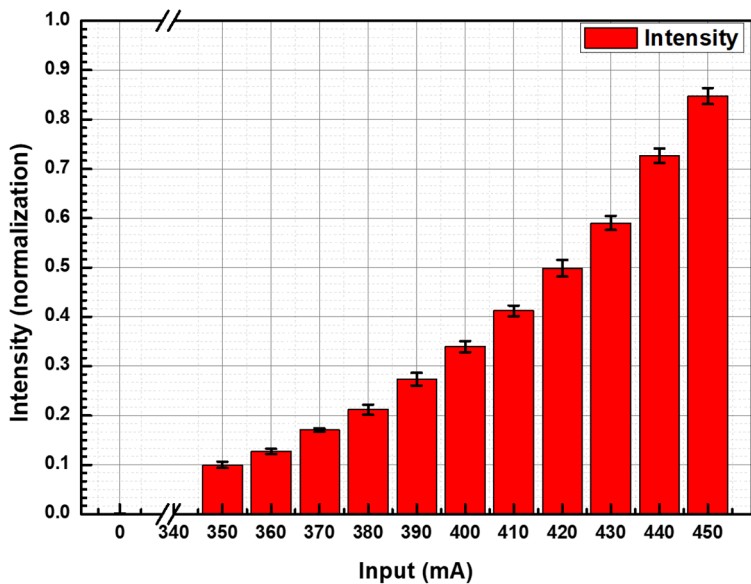

**Figure 4.** Intensity of the output pulse according to the magnitude of the current input to the OA.

The experimental results for the visibility of the MZ interferometer are shown in Figure 5. Figure 5 shows the results of the measurement while changing the driving voltage of the PM from −4.8 to 4.8. For both signal and decoy pulses, the $V_\pi$ value was 3.05 V, and the interference visibility was measured as 0.9437 and 0.9540, respectively. This means that the quantum bit error rate (QBER) for the signal and decoy pulses can be, at least, 2.82% and 2.29%, respectively, when QKD operates. Additionally, if considering the detector noise of dark count and afterpulse noise, the QBER may increase to 3.12% and 3.67%, respectively. The detector noises are likely to influence decoy pulses more than signal pulses. Interference can occur and QKD can be performed even if the proposed double amplification method changes the intensity of the pulse.

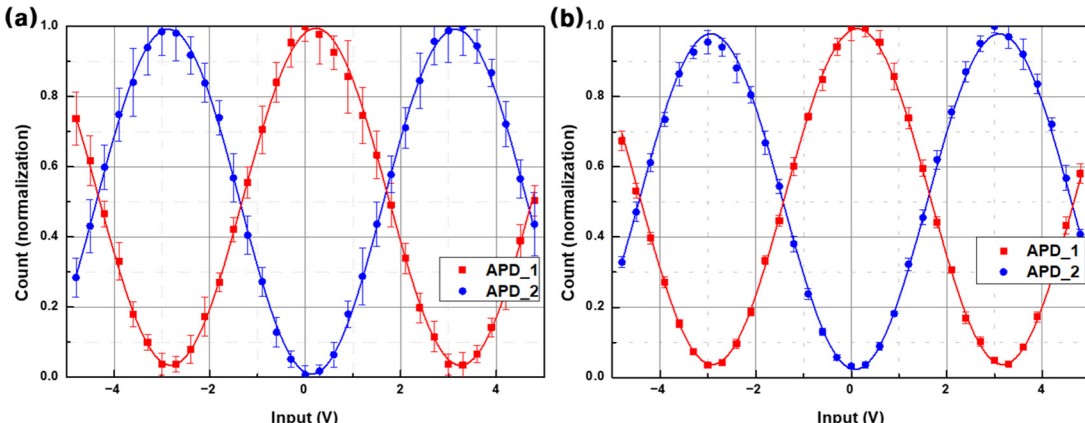

**Figure 5.** Interference visibility results in (**a**) signal and (**b**) decoy pulse. The measured counts of APD_1 (red) and APD_2 (blue) versus input voltage of PM are shown. It can be seen that the measured counts are well modeled with the sine function. The visibility of signal and decoy pulse are 0.9437 and 0.9540, respectively.

Finally, the experimental results during 26 h of operation of the PnP QKD system are shown in Figure 6. Signal and decoy pulses obtained average key generation rates of $5.37 \times 10^{-3}$ bit per pulse (b/p) and $1.037 \times 10^{-3}$ b/p, respectively, and QBER obtained averages of 3.35% and 3.95%, respectively. This result shows the stable key generation rate and QBER even after a long time of experimentation. Thus, it was shown that the proposed decoy generation could operate as stable, as we expected. The decrease in the key generation rate and increase in QBER are mainly due to the slight mismatch between the arrival of the light pulse and the gate timing of the single-photon detector. This mismatch may be due to changes in the laboratory environment and variations in device and equipment performance that may occur over the time of the experiment. This can be improved by adjusting the operating timing of the single-photon detector [24]. Because no automatic timing control system had been configured in our experiments yet, we adjusted optimal timing manually at 24 h. As soon as an adjustment of 1 nsec was made, the key generation rate and QBER were returned, as shown in Figure 6.

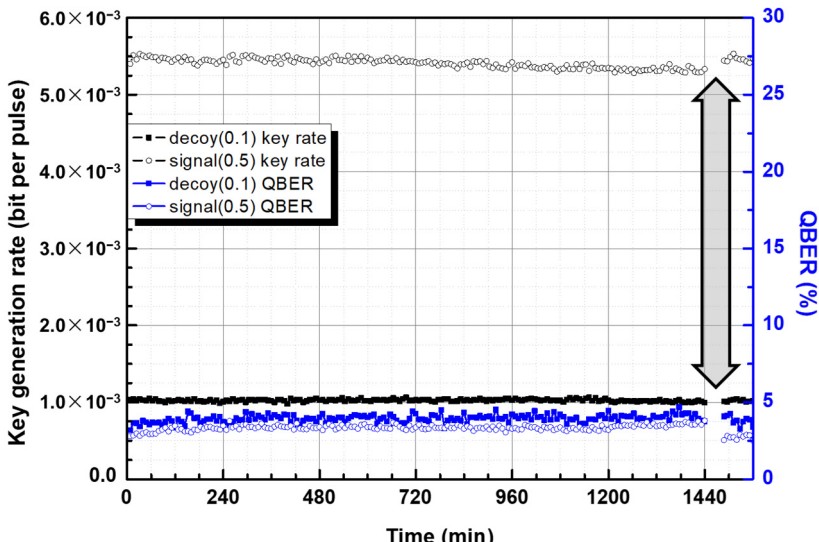

**Figure 6.** Key generation rate and QBER (signal and decoy) of the proposed PnP QKD. The experiment was conducted for a total of 26 h. After 24 h, the key generation rate and QBER were restored by manual control of the detector's gate timing. After that, measurements were taken for 2 more hours.

## 4. Conclusions

We proposed a PnP QKD system that generates decoy pulses using optical amplification. We showed that random polarization pulses are amplified without distortion using the double amplification method and verified this through actual experiments. Signal and decoy pulses were created by the open-loop control of the OA. The key was generated through PnP QKD for 26 h. Finally, the signal and decoy pulses obtained an average key generation rate of $5.37 \times 10^{-3}$ b/p and $1.037 \times 10^{-3}$ b/p, respectively, and QBER obtained averages of 3.35% and 3.95%, respectively. Our results show that the proposed generation method of decoy pulses using optical amplifiers works well. In the future, a more complete system including random modulation, timing control of the single-photon detector, the generation of a secret key after post-processing, etc., will be developed.

**Author Contributions:** Conceptualization, M.-K.W. and C.-H.P.; Formal analysis, M.-K.W.; Funding acquisition, S.K. and S.-W.H.; Investigation, M.-K.W. and C.-H.P.; Methodology, M.-K.W. and C.-H.P.; Project administration, S.K. and S.-W.H.; Supervision, S.-W.H.; Validation, C.-H.P.; Visualization, M.-K.W.; Writing—original draft, M.-K.W.; Writing—review and editing, S.K. and S.-W.H. All authors have read and agreed to the published version of the manuscript.

**Funding:** National Research Foundation of Korea (2019M3E4A1079777, 2021M1A2A2043892), Institute for Information and Communications Technology Promotion (2020-0-00972, 2020-0-00947), and the Korea Institute of Science and Technology (2E31531).

**Institutional Review Board Statement:** Not applicable.

**Informed Consent Statement:** Not applicable.

**Data Availability Statement:** Data underlying the results presented in this paper are not publicly available at this time but may be obtained from the authors upon reasonable request.

**Acknowledgments:** The authors wish to thank the anonymous reviewers for their valuable suggestions.

**Conflicts of Interest:** The authors declare no conflict of interest.

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
