# Peer review of "Generation of Decoy Signals Using Optical Amplifiers for a Plug-and-Play Quantum Key Distribution System"

_applsci, doi:10.3390/app12136491_

Round 1

Reviewer 1 Report

Dear Editor,

the manuscript entitled "Generation of decoy signals using optical amplifiers for a plug 2 and play quantum key distribution system" by Minki Woo and coworkers is a detail work reporting a novel approach to overcome some of the main issues that typical decoy-state protocols are still facing and that hinders their commercialization on a large scale.
Specifically, the authors propose to use an optical amplifier instead of an attenuator to improve the stability of the decoy signal. The submitted manuscript first illustrates the optical system developed by the authors and secondly reports experimental results showing the quality of their system.
Overall, the work is well structured and the discussion of both the novel experimental setup and of the experimental results is thorough and detailed.
I would therefore suggest the Editor to publish this manuscript on Applied Sciences after the minor revision listed below here are addressed by the authors.

1.When discussing the results reported in Figure 3, the authors say that the intensity ratio between |f> and |s> is very similar between the original and the amplification signals. Since this aspect is crucial in their work, it would be of great benefit for the quality and the soundness of this manuscript to show and indicate how reproducible such outstanding results are.

2.Labels of Figure 4 and Figure 5 are inverted.

3.Original Figure 4 (that should be instead Figure 5, according to the main text), reports two lines which are mentioned as "fitting" in the key line. However, no fitting procedure is discussed in the manuscript. The authors should amend this discrepancy.

4.Despite the experimental setup is described in Figure 1 and in the related discussion, the Method section should be reported anyway in the manuscript. At the moment, there is no Method section at all.

5.Since the authors correctly say that stability is crucial in QKD, they should briefly discuss and justify the small but still detectable drift they observe in Figure 6, with the QBER values going from 2.5/3 to around 4 and the signal key rate diminishing from 5.5e-3 to 5.25e-3.

6.Finally, I would like to invite the authors to read carefully the text before publication since a few typos can be spotted throughout it.

Reviewer 2 Report

The manuscript "Generation of decoy signals using optical amplifiers for a plug and play quantum key distribution system" solved the problem of generating different intensities with an arbitrary polarization input in a Plug and Play QKD system. It is somehow interesting, but I strongly advise the author to try their best to addres the following issues first.

1. In a decoy state protocol, phase randomization is required as well as the different decoy state generation. Please show it clearly on whether the   phase randomization is performed in the experiment, and how is that performed.

2. As discussed in the manuscript, the OA is saturated if the input is stronger than -25 dBm. The intensities of the signal and decoy states will be the same if the  OA is saturated. Please discuss if an loophole will be induced if Eve sends a stronger light to Alice, e.g., stronger than -25 dBm.

3. Please discuss if the mean photon number of the signal states will fluctuate if the intensity of the input light changes, e.g., when the loss of the channel changes.

4. It seems a static modulation for signal and decoy intensities are adopted in the experiment. In a decoy state based QKD experiment, the intensities of each pulse should be different, with a random number controlling the intensities.  Please make this point clear.

5. In the manuscript, the key rate of different states are directly recorded from the P&P system. When adopting the signal, decoy and vacuum state method, the gain of the different decoy intensities should be recorded, the yield and QBER of the single photon state should be calculated. These observed values need to be presented clearly and  final secure key rate based on the above variable should be calculated. 

6. Please discuss the working principle of an optical amplifier, and the reason the intensity is polarization independent with an OA modulation. Please discuss the effects of the noise induced by the OA.

7. It should be nice for the authors to make a more complete literature list on the most recent and important application of the decoy state method. For example, Y.-H. Zhou et al, Phys. Rev. A 93, 042324 (2016). Also, the SNS protocol Phys. Rev. A 98, 062323 (2018) which has been successfully applied in the experiments of Ref[4] and Ref[8] of the manuscript.

Reviewer 3 Report

The authors Woo et al. present a study on an optical pulse intensity modulation scheme suitable for decoy state quantum key distribution (QKD). The problem of using intensity modulators is their polarization sensitivity, requiring a careful analysis of the used apparatus and compensation. In the studied scenario, a plug-and-play QKD system this might not always be feasible, and therefore another scheme is desirable. Instead of the conventional intensity modulators, which often only attenuate a signal, the authors propose to amplify the signal instead. By double-passing the amplifier with a Faraday rotator in between, the authors achieve equal amplification gain for the polarization components. The gain is also variable by adjusting the current driving the amplifier. The authors also verify their scheme experimentally.

The idea of double-passing the polarization-sensitive component with a Faraday rotator is appealing and could be used for practical implementations of decoy state QKD. It raises the question if a “standard” intensity modulator could benefit from that scheme as well.

The manuscript, however, is unfortunately poorly written making it difficult to comprehend the work in detail. Before publication it needs significant improvements. This does not only concern the language, but also a lot more explanations are needed. The aspects that need to be worked on include (but are not limited to):

1) The authors state in the abstract and in the main text that decoy states are required to prevent the photon number splitting attack. This is not correct – if the mean photon number is sufficiently low, no information can be gained (the leaked information can be upper bounded and reduced to zero with privacy amplification). Decoy protocols, however, allow for higher mean photon rates and are therefore more efficient.

2) Most key words double essentially the title and are therefore useless for (internet) search engines.

3) Many abbreviations are not defined in the main text, e.g. QC is used in the abstract and only defined in the caption of figure 1.

4) Many sentences are not correct English, e.g. “… there are still conducting many researches about long-distance transmission…” or “… is one of the key issues of QKD system …” or “Essentially, quantum signal is fragile …” or “Single mode fiber, which is often used as a quantum channel, does not maintain pulse polarization” or “… the signal reaching OA must be large then -50dBm …”. There are many more examples.

5) Figure 1 is not explained at all. The only explanation is “The proposed structure was implemented as shown in Figure 1.” In the figure caption only the abbreviations are defined. It is not clear what the components do, unless one is an expert in the protocol. Only some functionalities are explained later in the main text.

6) In equation 6, the parameters A and B are rather the square-root of the probabilities, otherwise the initial state would not be normalized. I understand that the amplification is a non-unitary process and therefore not norm-conserving, but at least the initial state should be treated as a normalized state.

7) It should be explained why polarization maintaining fibers are not a possible solution.

8) What is the use of Attenuator_2? This should be in the explanation of the scheme.

9) Shouldn’t all light traveling back exit the circulator in Alice’s setup to PD_2? The authors claim the circulator would prevent the light from going through Attenuator_1 twice. I don’t understand how this is possible with the shown schematic.

10) There are random abbreviations (e.g. APN) that are not defined.

11) The authors simulate a quantum channel with a loss of 5 dB corresponding to 25 km. It would be good to explain why this is the case. I assume the authors took 0.2 dB/km which is the loss in the telecom C-band at 1550 nm. This raises the question of the used laser wavelength? Was this 1550 nm?

12) The authors mention e.g. 5.37E-3 as the key rate. A rate is not a dimensionless number, but something with the units 1/s. What the authors probably mean is the extractable secret information per signal.

13) The authors show a very similar interference visibility but then a moderate difference in the QBER (3.35% vs 3.95%) which is larger than what should be caused by the different interference visibility. What is the reason behind this? This difference will induce more required privacy amplification that in turn reduces the key rate.

Round 2

Reviewer 1 Report

1.Since the double amplification measurements shown on Figure 3 occurs on a sub-second scale (despite the labels on the x scale are barely readable, which makes the presentation of the data poor), it is surprising that, after my suggestion (Comment #1), they measure the reproducibility of their results over just 10 cycles.

2.Thanks to the auhtors for amending the typo concerning the label of the Figures.

3.Despite the authors removed the word "fitting" from the manuscript to comply with my comment #3, this makes the submitted work less precise. The sine curve is still in the graph, but neither in the main text, nor in the legend/caption it is reported what the blue and red lines are.

4.The Method section should report details on the equipment used to perform the experiments. The newly submitted version of the paper has not been improved on this side at all. Adding the "Method" title does not mean to have actually added the Method section. At this stage, the experiments are not reproducible by any other group.

5.The hypothesis that the drift in the data reported in Figure 6 might be due to a change in the temperature of the laboratory should be supprted by experimental evidence. Since measuring the temperature around room temperature is trivial, this should not require particular efforts by the authors. At this stage, the reply of the authors is inadequate.

6.I would like to thank the authors for having read the manuscript and amended some typos spotted throughout their work.

Reviewer 2 Report

I suggest acceptance.

Author Response

We appreciate your acceptance of my paper.

Reviewer 3 Report

The authors Woo et al. have revised and improved their manuscript. One of the key issues – that it was difficult to comprehend due to lacking explanations – is now resolved. The other key issue – that language improvements are required – remains to a certain degree, however. The authors state that they have used a professional English editing service but have not even addressed all examples that I specifically mentioned. For clarification, here is what was meant:

1) The sentence “To achieve a more prevalent technology, many researches are still conducting for long-distance transmission […] tests” should be either “To achieve a more prevalent technology, many researchers are still conducting long-distance transmission […] tests” or “To achieve a more prevalent technology, much research is still conducting on long-distance transmission […] tests”.

2) The sentence “Essentially, quantum signal is fragile to changes in the external environment…” should read “… any quantum signal…” or “…a quantum signal…” or “…quantum signals are…” such that the sentence reads smoothly.

3) “Single mode fiber, which is often used as a QC, does not maintain pulse polarization…” should read “Single mode fibers, which are often used…” or “A single mode fiber, which is often used…”

The authors should not see this as a complete list – there are more examples that I cannot explain all in detail. Even when copying the text into a modern Microsoft Word version or free online tools such as https://writer.com/grammar-checker/ I get suggestions to change these examples.

Further (minor) comments:

4) It should be the sum of the absolute squares of A and B that equals 1 (in Equation 1).

5) The figure caption of Figure 1 still does not explain the figure. A figure (with figure caption) alone should be sufficient to understand the figure. The experimental setup does not need to be explained in that detail as in the main text, but at least some explanation is needed.

6) The authors have calculated their QBER as QBER = 1 – V. Shouldn’t it be QBER = (1 – V) / 2 (see Equation 34 in Rev. Mod. Phys. 74, 145 (2002))? This also means that the QBER is not limited by the visibility, which requires further explanation in the text (maybe detector dark counts, misalignment, etc.).

7) The authors have improved their references, but there are still sometimes the DOIs mentioned.

Round 3

Reviewer 1 Report

Dear Editor,

the new version of this manuscript has been improved with respect to the previous two versions. Overall, my comments have been mostly addressed, despite I still have some minor concerns about a couple of them, as mentioned in detailed below here.

I would therefore recommend the Editor to publish the submitted manuscript after these minor issues have been amended.

Comment #1: First of all, I would like to thank the authors for having improved their work in compliance with my comment. Anyway, there is a discrepancy between the number of digits reported in the manuscript (up to 10^-4) and the error bars reported in Figure 3b. In the Figure, the error bars are clearly larger that 1*10^-4, thus indicating that the meaningful digits in the |s>/|f> ratio are definitely less than 4, as reported in the manuscript. This is the basic of an error analysis University course. Also, it is unusual and critical that in a physics paper on quantum information and quantum computing an experimental value is reported with no error. These trivial but fundamental aspects drastically lower the quality and the publishability of the manuscript submitted by the authors.

Comment #4: The authors have now included all the information required to reproduce the data reported in the manuscript. Despite this, the Methods section has not been organized as it is usually reported in most of the scientific manuscripts, i.e. as a separate section that gathers all the equipment details for all the experiments that will be reported in the manuscript. At the moment, the "Methods" are scattered throughout the discussion and, in one case, also in a Figure Caption (Fig.1), thus confusing the reader.

Comment #5: This comment has not been properly addressed. The authors still report in their newly submitted manuscript that the drift in their observation is due to a variation of the laboratory temperature with no experimental evidence that actually proves their statement. Also, in order to solve this issue, they say that they had to adjust the operation timing which, to the best of my knowledge, has almost no relation with the laboratory temperature. Thus, this aspect of the manuscript is still weak and needs improvements. On one side, if the authors think that the laboratory temperature is the real cause of the observed drift, they should prove it by showing experimental results that correlate the output of their system with temperature. On the other side, if the cause is due to some operation timing issues, the authors should provide deeper details on the cause of this change over time.

Reviewer 3 Report

The authors Woo et al. have revised and further improved their manuscript. There are still a few minor language issues, however, these can be fixed during type setting. I can therefore recommend the manuscript for publication now.

Author Response

Thank you for accepting our paper.